# NRF2 Mediates Cellular Resistance to Transformation, Radiation, and Inflammation in Mice

**DOI:** 10.3390/antiox11091649

**Published:** 2022-08-25

**Authors:** Dörthe Schaue, Ewa D. Micewicz, Josephine A. Ratikan, Keisuke S. Iwamoto, Erina Vlashi, J. Tyson McDonald, William H. McBride

**Affiliations:** 1Department of Radiation Oncology, David Geffen School of Medicine, University of California at Los Angeles, Los Angeles, CA 90095-1714, USA; 2Biotts S.A., Ul. Wrocławska 44C, 55-040 Bielany Wrocławskie, Poland; 3Department of Radiation Medicine, School of Medicine, Georgetown University, Washington, DC 20057, USA

**Keywords:** Nrf2, NF-κB, ionizing radiation, immune polarization

## Abstract

Nuclear factor erythroid 2-related factor 2 (NRF2) is recognized as a master transcription factor that regulates expression of numerous detoxifying and antioxidant cytoprotective genes. In fact, models of NRF2 deficiency indicate roles not only in redox regulation, but also in metabolism, inflammatory/autoimmune disease, cancer, and radioresistancy. Since ionizing radiation (IR) generates reactive oxygen species (ROS), it is not surprising it activates NRF2 pathways. However, unexpectedly, activation is often delayed for many days after the initial ROS burst. Here, we demonstrate that, as assayed by γ-H2AX staining, rapid DNA double strand break (DSB) formation by IR in primary mouse *Nrf2–/–* MEFs was not affected by loss of NRF2, and neither was DSB repair to any great extent. In spite of this, basal and IR-induced transformation was greatly enhanced, suggesting that NRF2 protects against late IR-induced genomic instability, at least in murine MEFs. Another possible IR- and NRF2-related event that could be altered is inflammation and NRF2 deficiency increased IR-induced NF-κB pro-inflammatory responses mostly late after exposure. The proclivity of NRF2 to restrain inflammation is also reflected in the reprogramming of tumor antigen-specific lymphocyte responses in mice where *Nrf2* k.o. switches Th2 responses to Th1 polarity. Delayed NRF2 responses to IR may be critical for the immune transition from prooxidant inflammation to antioxidant healing as well as in driving cellular radioresistance and survival. Targeting NRF2 to reprogram immunity could be of considerable therapeutic benefit in radiation and immunotherapy.

## 1. Introduction

Potentially biohazardous free radicals and non-radical intermediates, most notably reactive oxygen species (ROS), are generated during numerous physiological and pathological cellular processes, and cytoprotective, biological control strategies have evolved to coordinate and control such responses. A keystone in this process is nuclear factor erythroid 2–related factor 2 (NRF2; NFE2L2) that regulates batteries of antioxidant and detoxification genes [1]. Understanding NRF2 function presents both challenges and opportunities for intervention in multiple disease processes [2,3]. This paper explores how NRF2 controls cell transformation, radiation responses, inflammation and immunity. The reader is referred elsewhere for broader discussions [4,5,6,7,8].

NRF2 is a member of the cap‘n’collar-basic leucine zipper (CNC-bZIP) family of transcription factors that dimerize with small MAF proteins to bind antioxidant response elements (ARE) in promoter regions of NRF2 target genes [9]. While NRFs are the primary transcription factors to interact with AREs, others such as AP-1 may contribute [10]. Under physiological conditions NRF2 is maintained at very low levels. This is because it is bound by the Kelch-like ECH-associated protein 1 (KEAP1) through DLG and ETGE motifs in its Neh2 domain [11], leading to recognition by a Cullin 3-based E3 ubiquitin ligase and its rapid 26S proteasome degradation [12]. The primary control mechanism for NRF2 stabilization is through a redox-sensitive thiol switch on Keap1 [13] that releases NRF2 for nuclear translocation and transcription of ARE-dependent target genes [14,15]. However, other ways to both negatively and positively regulate NRF2 expression exist that vary in their dependency on cysteines, KEAP1, oxidative stress, and AREs [16]. Many questions remain as to how NRF2 responses differ depending on the stimuli, and the impact of factors such as the context, dose, time, cell type, species and strain, and sex [17].

The most complete data on NRF2 functions come from gene expression studies in *Nrf2–/–* and/or *Keap1–/–* mice and their derived mouse embryo fibroblasts (MEF) [1]. *Nrf2–/–* mice are viable, although they present pathologies consistent with persistent oxidative stress that shorten their lifespan [14]. *Keap1–/–* mice die postnatally within 3 weeks as constitutive *Nrf2* expression drives hyperkeratosis in the esophagus and forestomach [18]. While NRF2 is accepted as the master of oxidative/electrophilic stress responses, the literature points to roles for NRF2 in multiple biological processes [19,20]. Its clinical relevance in cancer biology and treatment is shown by NRF2 being dysregulated in many human cancers [21,22,23,24,25], especially where activating mutations, deletions, or gene amplifications affect NRF2/KEAP1 function to promote constitutive Nrf2 expression [15,26], with increased therapeutic resistance and poor outcomes [21,22,23,24,25]. The close communication between the NRF2/KEAP1 partners is nicely illustrated by their mutually exclusive mutation patterns in head and neck [27] and lung cancer [28].

One reason Nrf2 is empowered with so many, diverse functional roles is linked to its ability to reprogram cellular metabolism, which is critical for combatting oxidative stress. Activation of NRF2 increases glucose uptake by cells and redirects it into the pentose phosphate pathway (PPP) [15], while loss of NRF2 prevents this metabolic switch and the proper regulation of redox imbalance. However, in general the oxidative branch of the PPP contains NRF2 target genes that supply ribose 5-phosphate (R5P) for nucleotide synthesis and cell proliferation [29,30], which helps explain the association between NRF2 and poor cancer prognosis, and perhaps why growth factors can mediate cellular radioresistance [31]. It also provides the enzymes required for NADPH production, all of which are under NRF2 control [29,30,32]. NADPH protects against oxidative stress directly by scavenging ROS and indirectly by providing reducing equivalents for the redox cycling enzymes glutathione reductase (GR) and thioredoxin reductase 1 (TRXR1), whose synthesis is also under NRF2 control [15,33]. This involvement of the PPP and NADPH in redox homeostasis contrasts starkly with immune defense where the NADPH oxidase (NOX) complex generates large amounts of superoxide [34] and neutrophil extracellular traps (NETs) that localize and kill microorganisms [35]. Cells vary widely in their NOX isoform expression, but NRF2 can regulate production of ROS by at least some NADPH oxidases as well as by mitochondria [36], suggesting that NRF2 can mediate both the production and scavenging of ROS.

The involvement of NRF2 in metabolism indicates mechanisms by which it might alter responses to IR and affect the generation of immunity. IR generates ROS in cells rapidly by radiolysis of water, with later waves of ROS production coming from mitochondria leakage and NOX activation [37,38,39,40]. It is no surprise that IR stabilizes Nrf2 expression [4,41]. This would be expected to drive the PPP to enhance cellular radioresistance [42,43]. However, IR-induced NRF2 activation is not always rapid and is often delayed by many days [44], suggesting that initial ROS production is less relevant than later events. Indeed, while *Nrf2–/–* MEFs and mice are more radiosensitive than WT [41,44,45,46], immediate NRF2 responses induced by chemical agents [41,47] may not necessarily cytoprotect cells against IR [41]. This suggests that the oxidative stress induced by IR and downstream Nrf2 antioxidant production has many complexities but little is known about the roles of delayed NRF2 responses in the many biological processes thought to be under NRF2 control.

Here, we report that *Nrf2–/–* MEFs are more sensitive to spontaneous and to IR-induced transformation and that delayed genomic instability may be one process inhibited by late Nrf2 activity. Another possibility is that NRF2 responses are blocked by activation of redox-regulated, pro-inflammatory NF-κB family members that are known to have an inverse relationship with Nrf2 [48,49,50,51]. We present evidence that NRF2 deficiency increases IR-induced NF-κB pro-inflammatory responses most robustly late after exposure. Delayed NRF2 activation could be heavily involved in preventing or limiting inflammation and this is examined in models of Th1/Th2 immune polarization and raises questions as to the role of NRF2-mediated programs in the balance between pro-inflammatory, pro-oxidant responses and the antioxidant, tissue healing events that occur over time after IR exposure.

## 2. Materials and Methods

Male and female Nrf2 heterozygous mice were a kind gift from Dr. Andre Nel. C57BL/6 *Nrf2* wild-type (WT), heterozygous, or deficient (KO) mice were genotyped, housed, and bred in a gnotobiotic environment in an AALAC-accredited facility in the Department of Radiation Oncology, UCLA. The UCLA IACUC approved all experiments, which were performed in accordance with all local and national guidelines for the care and use of animals. Male mice, 10–16 weeks old, were used for experiments.

### 2.1. Cell Culture, In Vitro and In Vivo Irradiation

Unless noted, cells were cultured in DMEM (Invitrogen) with 10% fetal bovine serum (Omega Scientific, Tarzana, CA, USA) at 37 °C in humidified air containing 5% CO_2_. Cells were irradiated using a Shephard 137-cesium irradiator at a dose rate of approximately of 6 Gy/minute.

Mice were randomized to exclude litter effects and differences in body weight (26 ± 2 gms), and given whole body irradiation (WBI) without anesthesia using an AEC 137-cesium irradiator at a dose rate of approximately of 0.67 Gy/minute to their LD70/30 dose (8.2 Gy for WT and 7 Gy for *Nrf2–/–*, respectively). Weights were measured every other day between 1–4 wks after exposure as one measure of acute radiation sickness, as before [35]. If overt morbidity was observed, mice were euthanized in accordance with the approved protocol. 30-day survival was compared with and without 5 daily doses of 5 mg/kg s.c. BCN512 mitigator treatment (in 1% Cremophor EL) starting 24 h post exposure using Kaplan–Meier survival analysis and log rank statistics (*n* = 8 mice per group). Significance was assessed at the 5% level using SPSS v20 software (IBM SPSS Satistics, Armonk, NY, USA). Mitigator control animals received 1% Cremophor EL diluent only.

Dosimetry was performed by trained medical physicists using NIST-calibrated ion chamber, thermoluminescent dosimeters in a mouse phantom, and GafChromic™ EBT film (International Specialty Products, Wayne, NJ, USA) during experiments to ensure 95% homogeneity and reproducibility.

### 2.2. Foci Formation Assay

Primary MEFs expand in vitro until they exhibit density-dependent growth inhibition. In rare instances transformants arise that form mutilayered transformed foci that are more common after IR [52,53]. This transformation is typically associated with loss of contact inhibition, increased proliferation, and ability to form tumors in vivo. Focus formation is a standard assay to assess IR-induced transformation. Primary MEFs were prepared from euthanized embryos of pregnant *Nrf2–/–* and WT mice at about 13.5 days gestation by trypsin digestion and mechanical disruption and grown in T175 flasks for less than 3 passages. For experiments, MEFs were irradiated in T175 flasks, incubated for 2 days to allow damage fixation [52,53], and plated in 10 cm dishes. Media was changed weekly and after 3 weeks cells were fixed in 70% alcohol and stained in 0.5% crystal violet to visualize foci.

### 2.3. NF-κB Reporter Gene Assay 

Immortalized WT and *Nrf2–/–* MEFs were a kind gift from Dr. Wakabayashi [54]. They were grown in Iscove’s medium (Invitrogen) with 10% fetal bovine serum (Omega Scientific, Tarzana, CA, USA) and used to develop a stable NF-κB reporter gene assay using a NF-κB-luciferase lentiviral vector containing four tandemly arranged NF-κB response element upstream of a TATA promoter (BPS Systems Biosciences, San Diego, CA, USA) with a one step luciferase development. A lenti-viral GFP vector was used as a transduction control for transduction efficiency that could be evaluated by fluorescent microscopy or flow cytometry.

### 2.4. γ-H2AX Assay

A γ-H2AX flow cytometric assay was chosen for assessment of DNA double strand breaks (DSB) because of its sensitivity at basal levels as well as after low and high IR exposures [55,56]. This was performed as we described previously [56], except using primary *Nrf2–/–* and WT MEFs that were irradiated and incubated for various times before assay. Cells were washed in PBS, fixed in 99% ethanol, and stored for 1 h at 4 °C and overnight at −20 °C. Samples were pelleted and gently resuspended in 50 μL permeabilization buffer (Tris-buffered saline, 4% FBS, 0.1% Triton X-100) for 10 min. Cells were pelleted and 0.02 μg FITC-conjugated monoclonal anti-phospho-histone H2AX (ser139) (Millipore 16-202A clone JBW301) or control FITC-normal mouse Ig (Millipore 12-487) added in 100 μL permeabilization buffer. Samples with no antibody acted as an additional control. Tubes were shaken for 2 h in the dark before pelleting cells and resuspending them in Isoton (Becton Dickinson, San Jose, CA, USA) for flow cytometry using a FACScaliber (BD, San Jose, CA, USA) with 20,000 events collected per point and analyzed by CellQuest with data plotted by WinMDI 2.9.

### 2.5. Immune Profiling

Single cell suspensions were prepared from excised spleens after manual disassociation and red blood cell depletion with ACK (Lonza Walkersville, Inc., Walkersville, MD, USA) treatment in serum-free RPMI-1640. Clumps were removed by sieving through a 70 mm cell strainer. Cells were counted and used for flow cytometry or enzyme-linked immunospot (ELISPOT) assay (see below). For flow cytometry, 1–2 million cells resuspended in 20 μL goat serum were briefly incubated with 2 μL anti-mouse CD16/CD32 Fc-block (BD Pharmingen, clone 2.4G2) prior to 30 min incubation with a cocktail of antibodies in brilliant stain buffer (BD Horizon) on ice. The antibody cocktail for lymphocyte subset analysis comprised: V450-anti-mouse CD8 (clone 53-6.7, BD Horizon #560469), PerCp-Cy5.5-anti-mouse CD4 (clone RM4-5, BD Pharmingen #550954), PE-anti-mouse CD25 (clone PC61, BD Pharmingen #553866), FITC-anti-mouse Foxp3 (clone FJK-16se, Bioscience #11-5773-82), BV510-anti-mouse CD19 (clone 1D3, BD Horizon #562956). Myeloid cells markers were detected with: PE-Cy7-anti-mouse CD11b (clone M1/70, BD Pharmingen #552850), Alexa 700-anti-mouse Ly-6G (clone 1A8, BD Pharmingen #561236), PE-anti-mouse Ly-6C (clone AL-21, BD Pharmingen #560592), APC-anti-mouse F4/80 (clone BM8, Biolegend #123116), FITC-anti-mouse I-A[b] (clone AF6-120.1, BD Pharmingen #553551), APC-Cy7-anti-mouse CD11c (clone HL3, BD Pharmingen #561241), BV605-anti-mouse CD86 (clone GL1, BD Horizon #563055) with dead cell exclusion based on 7-AAD uptake (Biolegend). 100,000 cells were acquired using a 14-color LSRFortessa (BD Biosciences, Mountain View, CA, USA), and analysed by FlowJo (FlowJo LLC, Ashland, OR, USA) using single-marker expression levels apart from Tregs that were gated based on CD4+ CD25+ FoxP3+.

### 2.6. B16-OVA Tumor Model and Enzyme-Linked Immunospot (ELISPOT) Assay

We have published the B16-OVA tumor model previously [57]. In brief, B16-OVA was a gift from Dr. Economou, University of California at Los Angeles and cultured in RPMI-1640 media (Mediatech, Hernden, VA, USA) with 10% fetal bovine serum (Sigma-Aldrich, St. Louis, MO, USA), 10,000 IU penicillin, 10,000 μg/mL streptomycin, 25 μg/mL amphotericin (Mediatech), 0.05 mM 2-mercaptoethanol (Sigma), and kept under 0.4 mg/mL G418 selection (Research Products International Corp., Mt. Prospect, IL, USA). A vaccine of B16-OVA cells was made by lethally irradiating them (25 Gy) and injecting 10^7^ cells, or saline, i.p. into *Nrf2–*/*–* or WT mice. Seven days later, spleens were harvested and OVA-specific IL-4 and IFN-γ producing T cells enumerated by ELISPOT assay by plating 2 × 10^5^/well for 48 h in anti-IL4- or anti-IFNγ-coated (BD Pharmingen, Franklin Lakes, NJ, USA) MultiScreen-HA plates (Millipore Corp, Billerica, MA, USA). Cells secreting IFNγ or IL-4 were detected using the corresponding biotinylated anti-IL-4 or anti-IFNγ antibodies (BD Pharmingen) and 200× diluted horseradish peroxidase avidin D (Vector Laboratories, Burlingame, CA, USA), with spots being developed in the presence of 0.4 mg/mL 3-amino-9-ethyla-carbazole substrate (AEC, Sigma) and counted using an ImmunoSpot Image Analyzer (Cellular Technology Ltd., Cleveland, OH, USA).

## 3. Results

### 3.1. Nrf2 and IR Exposure

*Nrf2–/–* mice are sensitive to chemical and radiation carcinogenesis [16,58,59], suggesting that NRF2 is protective against carcinogenesis, even though Nrf2 overexpression indicates a poor cancer prognosis. This prompted us to perform a focus assay for transformants WT and *Nrf2–/–* primary MEFs. *Nrf2*-deficient MEFs had a dramatically increased basal number of transformed foci over WT that had almost none. This difference was greatly amplified by IR exposure (2 or 4 Gy) (Figure 1A), in keeping with NRF2 protecting cells against carcinogenesis.

We had previously shown that, unlike WT MEFs, *Nrf2–/–* MEFs have high basal levels of ROS that increase more dramatically after IR than do WT MEFs [41]. This may relate to their 50% lower GSH levels, which others have also reported [36,60]. The higher basal and IR-induced ROS in *Nrf2–/–* MEFs might be expected to enhance DNA double strand (DSB) break formation and increased radiosensitivity. We therefore used a γ-H2AX assay to assess DNA DSB formation and repair. In fact, there was no difference in basal or IR-induced DSB formation between *Nrf2–/–* and WT MEFs (Figure 1B) even over a dose-dependent range of DNA DSB formation. DNA DSB repair was examined by γ-H2AX assay at various times after 8 Gy exposure from 0.5 to 24 h. As repair proceeded there was only a marginally slower repair in *Nrf2–/–* MEFs at 1 and 2 h. This could possibly reflect a small increase in complex lesions, although any difference in residual damage had disappeared by 4 h (Figure 1C) and 24 h (not shown).

*Nrf2–/–* MEFs and mice are more radiosensitive than WT [41,44,45], although the magnitude of the effect has rarely been examined in detail. Previously, we reported on survival of WT and *Nrf2–/–* gnotobiotic mice given a range of WBI doses where we performed the appropriate SAS probit analysis for radiation dose-related lethality. We found that the LD50/30 for hematopoietic acute radiation syndrome (hARS) between day 10–30 was reduced to 7.0 Gy in *Nrf2–/–*, as opposed to 8.2 Gy for WT controls [46]. These data allowed us to give isoeffective survival WBI doses to WT and *Nrf2–/–* mice and test if the activity of one of our most powerful 4-nitrophenylsulfonamide hARS mitigators, BCN512, [46] was Nrf2-dependent. BCN512 was inactive in *Nrf2–/–* mice, even though they were given lower WBI doses, while it almost completely protected WT mice from hARS lethality when given daily for 5 days starting 24 h after WBI (Figure 1D). Literature reports have also suggested a NRF2 dependency of certain other hARS mitigators [44,61,62,63,64]. In the case of BCN512 there is evidence that it acts through enhancing myelopoiesis after WBI [46], a process that involves cytokines and chemokines, which may be under NRF2 control.

### 3.2. Nrf2 Control of Inflammation and Immunity

IR can signal “danger” to activate innate and adaptive immune systems [65,66]. This involves NF-κB- and TLR-related pro-inflammatory cytokines for which NRF2 is a negative regulator [37]. This explains why *Nrf2–/–* mice display greater than normal basal and pathogen-induced pro-inflammatory states [48,49,50,51]. To examine the crosstalk between NRF2 and NF-κB, we transduced *Nrf2–/–* MEFs with a lentiviral NF-κB luciferase reporter. As expected, after 8 h these showed more pronounced responses to LPS and TNF-α than WT MEFs did (Figure 2A). However, neither MEF strain displayed a significant response to 8 Gy IR exposure within 8 or 24 h (Figure 2B), while both did at 6 and 9 days although in all cases NF-κB levels were higher and responses superior in *Nrf2–/–* compared to WT cells (Figure 2C). This is in line with NRF2 being a brake on NF-κB-induced inflammation. The lack of an acute NF-κB response to IR by WT cells is surprising as such responses have been reported previously. However, cell type may be important as may be the dose, which was often higher than was used here [67,68].

Immune defects have been found in *Nrf2–/–* mice by others. In our hands, the spleens of untreated *Nrf2–/–* mice had 2–3-fold more splenocytes than those from WT C57Bl/6 mice, which was not associated with any obvious splenomegaly (Figure 3A). The most notable differences in *Nrf2–/–* spleen compared to WT spleens were a larger B cell compartment (52.8% vs. 35.8% of total, Figure 3B), and an elevated CD4:CD8 lymphocyte ratio (2 vs. 1.1, Figure 3C), largely due to a decrease in CD8+ T cell representation which was 45% of normal (10.9% vs. 24.1%). There were also 32% fewer splenic Tregs in the absence of NRF2 (4.6% vs. 6.8%) while the myeloid compartment increased by almost 25% (11.3% vs. 9.2%).

In mice, 30 h after LD70/30 doses of WBI (7.25 Gy for *Nrf2–/–* and 8.51 Gy for WT) splenocyte counts, splenic B cells, and CD8+ T cells were dramatically decreased in keeping with their known radiosensitivities (Figure 3D). CD4+ T cells tend to be more radioresistant than their CD8 counterparts resulting in increases in the CD4:CD8 ratio in both strains after IR. This also happens when splenocytes are irradiated in vitro with 2 Gy (Figure 3D). The myeloid compartment was dramatically enhanced after WBI and there was good concordance between Nrf2-deficient and WT mice with major contributions from immature CD11b^hi^Ly6G^+^C^+^ cells. Differences in radiosensitivity of lymphocytes from *Nrf2–/–* mice and WT mice may be due to different GSH levels, which are known to moderate the sensitivity of T cells to apoptosis [69].

We used a B16-OVA cancer model in vivo to determine if systemic antigen-specific T cell functional polarity was altered in mice by loss of *Nrf2–/–*. A vaccine consisting of 10^7^ lethally irradiated B16-OVA cells was injected i.p. into WT or *Nrf2–/–* mice and IL-4 and IFN-γ responses assayed by ELISPOT using splenocytes harvested 7 days later (Figure 3E). IL-4 OVA-specific Th2 responses were decreased in *Nrf2–/–* mice compared with controls whereas IFN-γ responses were increased. These experiments indicate that NRF2 can control T cell antigen-specific polarization to drive Th2 responses, which we have also observed in vitro using direct CD3/CD28 T cell stimulation [3].

## 4. Discussion

The known consequences of *Nrf2* deficiency are multiple and include redox dysregulation, metabolic disease, cancer predisposition, sensitivity to IR, multiorgan inflammatory lesions, and autoimmunity [51,70]. It is worth noting that in many respects such a fundamental loss of control of redox, PPP activation, DNA damage response, and cell cycle, Nrf2 deficiency mirrors loss of the autosomal recessive ataxia telangiectasia gene (ATM) that makes patients and mice cancer prone, radiosensitive, and display a variety of inflammatory manifestations [71,72,73,74,75]. In fact, ATM and NRF2 pathways also crosstalk, in that NRF2 can activate ATM and ATR to cause G2 cell cycle arrest [76] while NRF2 inhibition represses ATM and ATR expression leading to aberrant DNA damage responses [77], and cells from A-T patients fail to upregulate Nrf2 [78].

Our finding that *Nrf2–/–* MEFs have increased basal and radiation-induced production of transformants is also seen in hemizygous ATM cells [79]. Interestingly, *Nrf2–/–* MEFs had no increase in the incidence of basal or IR-induced DNA DSB formation, and their repair was only slightly and transiently slowed; despite them having greatly enhanced basal and IR-induced ROS and low glutathione levels [41]. Clearly, higher ROS levels do not necessarily translate in more DNA DSBs. This is also true for highly radiosensitive A-T cells in which IR-induced DSB formation is normal and repair defective for only a small fraction of cells [73]. Jayakumar et al. [80] have proposed that NRF2 facilitates ROS-independent homologous recombination repair by direct interaction with DNA DSBs, which could explain the transiently slowed DSB repair we observed in *Nrf2–/–* MEFs. Our finding of a higher propensity for transformation in primary *Nrf2–/–*MEFs may result from these minor DNA repair abnormalities but be sufficient to lead to later chromosomal and genomic instability, as others have proposed linking transformation to carcinogenesis [81,82], although we have no direct evidence that this is the case. The role of NRF2 in inhibiting IR-induced transformation may also mechanistically relate to delayed induction of NRF2 and NRF2-dependent antioxidant target gene expression we reported earlier [41].

Why rapid ROS does not activate NRF2 is uncertain. ROS compartmentalization may be involved but a more appealing mechanism is coordinated, mutually antagonistic interactions between NRF2 and NF-κB, where IR-induced NF-κB blocks NRF2 activation. Kobayashi et al. [83] found that NRF2 interferes with upregulation of lipopolysaccharide-induced pro-inflammatory genes in macrophages by a mechanism that was ARE and ROS independent and suggested that transcription regulation of NRF2 was more important than redox switches. The appeal of this dualism between NRF2 and NF-κB lies in the consideration that the NRF2-PPP axis might mediate the transition from acute NF-κB-driven inflammation to Nrf2-driven healing, illustrated in the graphical abstract. The delay in NRF2 induction may allow time for elimination of pathogens or for DAMPs to generate responses [84,85,86,87]. We considered the possibility that IR-induced NF-κB might delay NRF2 activation, but at least at the radiation doses examined, optimum NF-κB responses were as delayed as NRF2 responses although we did observe some low level, acute responses. We have published extensively on IR-induced acute and delayed proinflammatory cytokine production [88,89,90] and NF-κB responses can certainly be induced rapidly by IR, but in most cases require high radiation doses [68]. Further studies are needed to explore this dualism between NF-κB and NRF2, but at this time the failure of high ROS levels to induce NRF2 in many cell types remains a conundrum. It seems the hypothesis of oxidative stress driving NRF2-mediated antioxidant responses is oversimplified and that NRF2 can march to the beat of many different drums. The relationship between delayed NRF2 responses and prevention of chromosomal instability also requires further study, as does the potential link of NRF2 to IR-induced “danger” signals, be they from DNA fragments, ROS, or pro-inflammatory cytokines. Certainly, the TLR activated positive feedback loop that is generated seems best placed in terms of activating inflammation and the immune system after IR. *Nrf2* deficiency would enhance these processes, as does loss of ATM [75].

It should be noted that not all IR-induced NRF2 responses are delayed, and this may be cell type-dependent. We have shown that murine bone-marrow-derived macrophages respond to IR through a surprisingly small number of sensors and signaling pathways; notably, IR-induced ROS act through NRF2 to regulate five radiation-induced early-response genes–*Slc7a11*, *Chac1*, *Trib3*, *Slc7a1*, and *Slc7a5*–all associated directly or indirectly with GSH regulation or activity [91]. Human hematopoietic stem cells (HSCs) can respond, though transiently, to IR (2 Gy) within 6 h with NRF2 target gene expression [45]. Even doses as low as 0.02 Gy have been shown to induce ROS rapidly but transiently to stabilize NRF2 in long-term HSCs, causing them to enter the cell cycle and differentiate, exhausting their reconstituting capacity [92]. This was associated with persistent oxidative stress and myeloid skewing.

Systemic alterations in the steady-state immunohematopoietic system of *Nrf2–/–* mice have been frequently, but variably, reported. We found an approximately 2-fold increase in numbers of splenocytes in *Nrf2–/–* gnotobiotic mice. The increase could be accounted for largely by B and myeloid cells, while CD8+ and Treg cells were decreased. Others have found enhanced proliferative ability of CD4+ lymphocytes [93] which could contribute to the increased number. In bone marrow, increased numbers of hematopoietic progenitors at the expense of HSC quiescence and self-renewal have been reported to lead to HSC exhaustion [94], which is further accentuated by low dose IR [92]. In fact, HSC exhaustion associated with high ROS levels occurs in *Atm*–/– as well as *Nrf2–/–* mice [95]. These immuno-hematopoietic differences associated with loss of *Nrf2* may vary with age, sex and strain, but a reasonable hypothesis is that they are related to decreased GSH levels and redox imbalance and may take different manifestations in different immune cell type and by extension link to their relative radiosensitivities [96]. As we [97], and others [98], have shown, *Nrf2* deficiency is accompanied by myeloid skewing. Myeloid-derived suppressor cells (MDSCs) are also elevated [99]. Indeed, the persistent myeloid skewing we noted after WBI [100] is most likely due to persistent ROS production and a failure in redox control.

The ability of NRF2 to control ROS levels and redox homeostasis in general is consistent with it exerting control over pro-inflammatory responses that also generate ROS. Not surprisingly, we found previously that loss of NRF2 resulted in Th1 immune polarization after stimulation of splenocytes by anti-CD3/CD28 in vitro [3]. Here, we extend this observation to in vivo antigen-specific OVA responses to lethally irradiated B16-OVA cells given as an in vivo vaccine. Again, NRF2 appears to inhibit Th1 and promote Th2 anti-inflammatory responses, which has also been observed in response to the NRF2 inducer tBHQ [101]. This is also consistent with Murata et al. who proposed that two classes of macrophages with low versus high intracellular GSH content drive CD4+ Th1/Th2 differentiation [102]. Notwithstanding some shortcomings in *Nrf2* knockout models and the complexity between baseline versus inducible antioxidant cytoprotection, the overarching concept that emerges is one where redox through GSH and TRX regulates inflammatory phenotypes in immune and non-immune cells. IR induces expression of pro-inflammatory cytokines such as TNF-α and IL-1 [65] that, along with cytoplasmic and released nuclear and mitochondrial DNA fragments and other IR damage-associated molecular patterns (DAMPs), act as “danger” signals for immune activation. Our immune polarization studies further suggest that targeting NRF2, especially in those tumors that constitutively express it, is an appealing way to enhance the therapeutic benefit of radiation and immunotherapy. The timing of such NRF2 responses may be especially relevant for fractionated IR treatments in the clinic and for the induction of IR-induced tumor immunity, which is being considered as a means to expand the use of immune checkpoint inhibitors in cancer treatment [3,103,104].

## Figures and Tables

**Figure 1 antioxidants-11-01649-f001:**
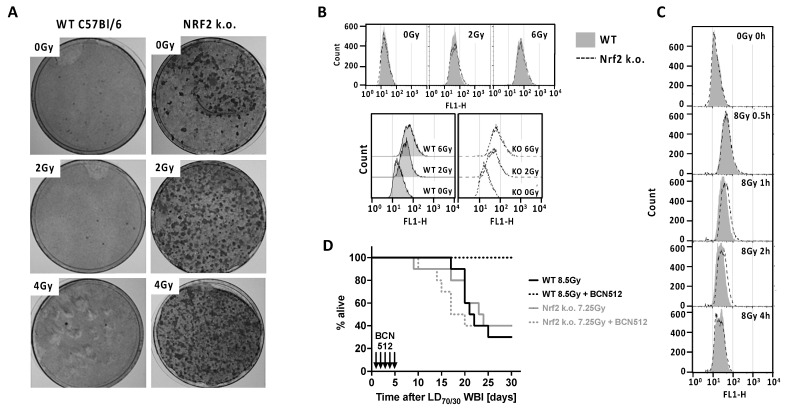
(**A**) Focus formation by primary MEFs from embryos of *Nrf2* knock out is considerably greater than in WT C57Bl/6 mice, a difference that was amplified by IR. MEFs were grown for <3 passages before irradiation with 0, 2, or 4 Gy. After 2 days culture to allow for DNA damage fixation, cells were plated in 10 cm dishes and after 3 weeks cells were fixed and stained to visualize foci. Shown are representative plates of *n* = 3 replicates that was reproducibly repeated in a second experiment. (**B**) A γ–H2AX flow cytometry assay was used to assess DNA DSBs 1 h after irradiation in *n* = 20,000 cells/sample and repeated once. IR increased γ–H2AX in a dose-dependent fashion, but there was no difference between *Nrf2–/–* and WT MEFs in levels of basal or IR-induced (2 or 6 Gy) DNA DSBs. (**C**) Repair of DSBs was assessed by incubating 8 Gy irradiated cells for varying time periods before γ–H2AX assay. Repair was slightly slowed in *Nrf2* k.o. MEFs compared to WT at 1 and 2 h, but this was transient and there was no difference at 4 h and thereafter (not shown). Data are representative histograms of *n* = 20,000 cells/sample. (**D**) *Nrf2* k.o. and WT C57Bl/6 mice were given WBI at LD70/30 values estimated from probit analysis. Twenty-four hours later the mitigator BCN512 was injected s.c. at 5 mg/kg in 1% Cremophor EL and hARS lethality assessed over 30 days. BCN512 was effective only in WT mice indicating the mitigator activity was NRF2-dependent. *n* = 10 mice/group.

**Figure 2 antioxidants-11-01649-f002:**
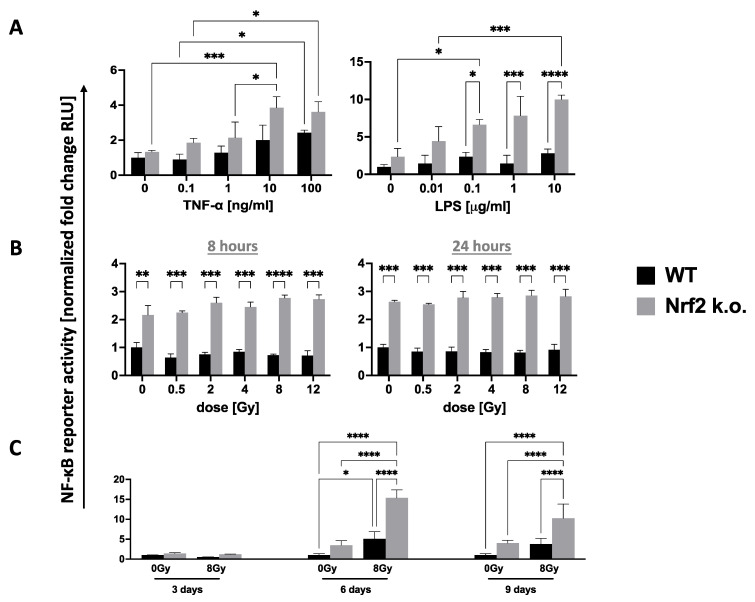
An NF-κB-luciferase reporter gene was transfected into long-term *Nrf2–/–* and WT MEFs and stimulated with (**A**) TNF or LPS, or (**B**) IR at indicated doses. TNF-α and LPS gave positive dose-dependent responses after 8 h (**A**) but not after IR at 8 or 24 h (**B**), which required several days (**C**). In general, responses were more pronounced in *Nrf2–/–* MEFs. Data are mean ± s.d. of *n* = 3–4; representative of 2 experiments. * *p* < 0.05, ** *p* < 0.01, *** *p* < 0.001, **** *p* < 0.0001.

**Figure 3 antioxidants-11-01649-f003:**
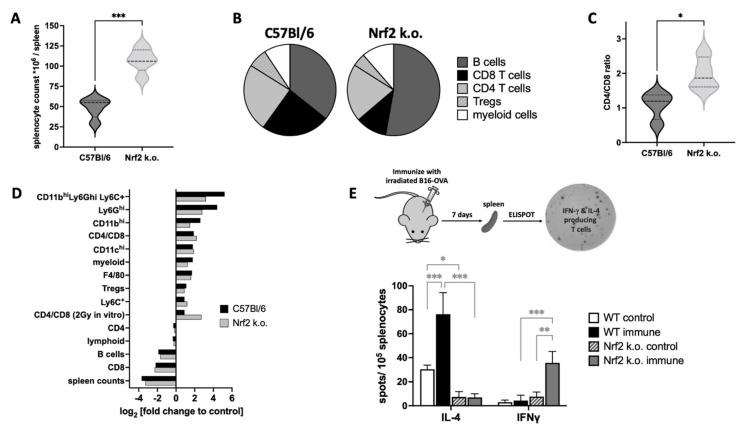
(**A**) *Nrf2* k.o. mice had 2-fold more splenocytes than WT C57Bl/6 controls (data are the median and interquartile range of splenocyte counts of *n* = 5 spleens/strain). (**B**) Most of the increase in splenocytes in *Nrf2* k.o. mice was due to a rise in B cells, with lesser increases in myeloid cells and decreases in CD8+ T cells and Tregs. (**C**) The decrease in CD8+ splenocytes were the main reason for increases in the median CD4:CD8 ratio (*n* = 4). (**D**) *Nrf2–/–* CD8+ cells were more sensitive to 2 Gy in vitro irradiation than WT resulting in a greatly increased CD4:CD8 ratio, and spleens harvested 30 h after LD70/30 WBI had decreases in both CD8+ T cells and B cells while myeloid cells, especially more immature CD11b^hi^Ly6G^+^C^+^ cells increased. Data are log2-fold changes (irradiated vs. control) in 10^5^ cells from *n* = 3 pooled spleen/group. (**E**) A vaccine of 10^7^ lethally irradiated (25 Gy) B16-OVA cells was injected i.p. into WT or *Nrf2–/–* mice and IL-4 and IFN-γ responses in splenocytes assayed by ELISPOT 7 days later. WT immune mice showed strong IL-4 responses while *Nrf2* k.o. immune mice made strong IFN-γ responses, suggesting that NRF2 controls immune Th cell polarization. Data are mean ± s.d. of *n* = 3; repeated once. * *p* < 0.05, ** *p* < 0.01, *** *p* < 0.001.

## Data Availability

Data supporting reported results can be obtained from the authors who generated the data or accessed through contacting the corresponding author. The data are not publicly available due to privacy. No publicly archived datasets were analyzed or generated during the study.

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
