# Peer review of "NRF2 Mediates Cellular Resistance to Transformation, Radiation, and Inflammation in Mice"

_antioxidants, 2022, doi:10.3390/antiox11091649_

Round 1
Reviewer 1 Report (Previous Reviewer 1)
Thank you for your revision of the article “NRF2 Mediates Cellular Resistance to Transformation, Radiation, and Inflammation”. The current title is improved now. I find the majority of revisions satisfactory. However, the following points need to be further addressed to improve the quality of this report.
Major comments:
1. There is no detailed method description of Figure 1D and Figure 3A,B,C,D experiments. Please supplement them in the Materials and Methods part, specifying what doses applied, how to perform probit analysis, and how to separate and count different kinds of cells.
2. Please perform statistical analysis of results in Figure 2.
3. Please include the number of experimental replicates and sample sizes in the figure legends.
4. The text needs careful editing for English language.
Minor comments:
Page 2, line 45: “Nfe2/2” should be “NFE2L2”
Page 4, line 130: “dys” should be “days”
Page 5, line 173: “NRF-/- MEFs” should be “Nrf2-/- MEFs”
Page 8, line 225: Please remove repetitions of “also”
Page 8, line 231: “them” should be “they”
Page 15: Reference 82 cannot be found in PubMed
Author Response
Please see the attachment.

Reviewer 2 Report (New Reviewer)
In the manuscript “NRF2 mediates cellular resistance to transformation, radiation and inflammation” Schaue et al. investigated the role of NRF2 after radiation of MEF’s and mice and in inflammation.
Although most of the work is sound, most studies lack proper controls and biological repeats in order to confirm the effects observed. It is very unclear how many times experiments were repeated and if biological repeats were included, or if these were only observations of singe experiments. This kind of information, including the statistics applied, must be incorporated in the materials and methods. In general, the manuscript could be written in a clearer way. Especially the description of the results could be improved to provide more support for a better structure and explanation of the connection between the studies performed.
1) The title is much to general and very a-specific. At the end, only experiments in mice spleen and primary MEFs were performed, so it is unclear how the results could be translated to e.g. human NRF2. This must be made clear both in the title and the abstract.
2) Was there any ethical approval for performing the mouse experiments?
3) The repair of double strand breaks in was detected via gamma-H2AX flow cytometry in radiated NRF2 -/- MEFs and displayed in Fig 2B and C. However, only single flow cytometry observations are shown, and in the text observations are described as “marginally slower repair” and “could possible reflect a small increase”. Correct investigation of these effects on DSB include doing proper biological controls, quantify the effects and perform statistics in order to substantiate the observations describe now. Also a proper positive control like using H2O2 for the detectability of DSB by this assay should be included.
4) The correct experimental setting of the mouse experiments with BCN512 should be included in materials and methods
5) Figure 3 D lacks biological repeats, and the application of correct statistics.
6) Overall, the discussion would profit from a scheme which clearly describes the different roles of NRF2 in inflammation and antioxidant effects, this would help to structure the discussion and to summarize the observations made in a better way.
Author Response
Please see the attachment.

Reviewer 3 Report (New Reviewer)
In this study, the authors demonstrated that the role of NRF2 on MEF transformation and inflammation triggered by radiation & pro-inflammatory cytokines. The manuscript has novel findings and several points which are deserved readers' attention.
However, the tile of this manuscript is vague. There are multiple inducers to transform MEF, but authors just transformed MEF via passaging. Similarly, cell inflammation can be caused by pro-inflammatory cytokines, such as TNF-alpha after radiation. Therefore, they authors should change the tile of this manuscript appropriately.
Round 2
Reviewer 2 Report (New Reviewer)
I have no further comments and this manuscrip is in my opinion now acceptable for publication.
This manuscript is a resubmission of an earlier submission. The following is a list of the peer review reports and author responses from that submission.
Round 1
Reviewer 1 Report
The manuscript submitted by Schaue et al. is aimed to link radiation and immunity based on NRF2. The introduction like a mini-review is interesting for me. However, I cannot appreciate the research design and the experiment results, which are not convincing. The method description is too rough to understand. There are some interesting findings, such as BCN512 mitigates hARS depending on Nrf2, but no explanation mentioned. In my opinion the manuscript is not acceptable for publication.
General comments:
The aims of the study are interesting and the questions raised in the manuscript deserve to be investigated. I suggest rewriting the manuscript more logically and clearly.
Specific comments:
1) Figure 3 is missing so that I cannot evaluate this part of results.
2) The title is not specific. It looks like a review title.
3) Some methods are missing, such as the method used in Fig 2.
4) The quality of Fig1A,B,C is not good enough. I can't see the horizontal and vertical coordinates clearly.
5) For Fig1A, I don’t understand the difference between focus assay and clonogenic assay and what transformants (Page 6, line 227) are. Why radiosensitive Nrf2 -/- MEFs show more foci even with IR treatment?
6) γH2AX foci are recognized as DSBs marker. Usually, counting γH2AX foci by immunofluorescence is performed to assess the number of DSBs. Flow cytometry may be applicable, but the description of γ-H2AX assay (Page 5, line 197-202) is not complete, even not mention how to incubate with γH2AX antibody and what kind of γH2AX antibody was used.
Reviewer 2 Report
This reviewer cannot review the entire manuscript because critical data of figure 3 are not included in the text.
1. Authors have previously reported that Nrf2-deficient MEFs are sensitive to radiation compared to wild type MEFs by using a colony assay. However, colony formation was not detectable and radiation effect was not seen in non-irradiated WT MEFs in this study. Is this a clonogenic assay for cell survival or soft agar colony formation (SACF) assay for detecting transformed cells? Authors should describe the detail methods for this assay.
2. Since the results of ROS measurement in Nrf2-deficient MEFs were reported in other paper, authors should not include ROS experiment in results section (line 231-240).
3. Indicate the time after irradiation for gamma-H2AX assay in Figure 1B. The reproducibility for FACS analysis is unclear in Figure 1B and 1C.
4. This reviewer cannot review the part of text (line274-299) because authors did not indicate Figure 3 in the text.